environmental chemistry/analytical chemistry/ materials science

pharmaceutical, removal, graphene, sulfamethoxazole, acetaminophen, environmental samples

**Author for correspondence:**

This article has been edited by the Royal Society of Chemistry, including the commissioning, peer review process and editorial aspects up to the point of acceptance.

# Efficient removal of pharmaceuticals from water using graphene nanoplatelets as adsorbent

Fatin Ahza Rosli[1], Haslina Ahmad[1,2], Khairulazhar Jumbri[4], Abdul Halim Abdullah[1,3], Sazlinda Kamaruzaman[1] and Nor Ain Fathihah Abdullah[4]

[1]Department of Chemistry, Faculty of Science, Universiti Putra Malaysia, 43400 Serdang, Selangor, Malaysia
[2]Integrated Chemical Biophysics Research Centre, Faculty of Science, Universiti Putra Malaysia, 43400 Serdang, Selangor, Malaysia
[3]Institute of Advanced Technology, Universiti Putra Malaysia, 43400 Serdang, Selangor, Malaysia
[4]Department of Fundamental and Applied Sciences, Faculty of Science and Information Technology, Universiti Teknologi PETRONAS, 32610 Seri Iskandar, Perak, Malaysia

FAR, 0000-0001-9849-7286; HA, 0000-0002-0796-414X; KJ, 0000-0003-3345-6453; AHA, 0000-0002-9313-1573; SK, 0000-0001-6299-8767

Recently, pharmaceutical pollutants in water have emerged as a global concern as they give threat to human health and the environment. In this study, graphene nanoplatelets (GNPs) were used to efficiently remove antibiotics sulfamethoxazole (SMX) and analgesic acetaminophen (ACM) as pharmaceutical pollutants from water by an adsorption process. GNPs; C750, C300, M15 and M5 were characterized by high-resolution transmission electron microscopy, Raman spectroscopy, X-ray diffraction and Brunauer–Emmett–Teller. The effects of several parameters viz. solution pH, adsorbent amount, initial concentration and contact time were studied. The parameters were optimized by a batch adsorption process and the maximum removal efficiency for both pharmaceuticals was 99%. The adsorption kinetics and isotherms models were employed, and the experimental data were best analysed with pseudo-second kinetic and Langmuir isotherm with maximum adsorption capacity ($Q_m$) of 210.08 mg g$^{-1}$ for SMX and 56.21 mg g$^{-1}$ for ACM. A regeneration study was applied using different eluents; 5% ethanol-deionized water 0.005 M NaOH and HCl. GNP C300 was able to remove most of both pollutants from environmental water samples. Molecular docking was used

to simulate the adsorption mechanism of GNP C300 towards SMX and ACM with a free binding energy of $-7.54$ kcal mol$^{-1}$ and $-5.29$ kcal mol$^{-1}$, respectively, which revealed adsorption occurred spontaneously.

# 1. Introduction

Water pollutants found in the environment can be divided into several categories viz. heavy metals, dyes, pesticides, pharmaceuticals and personal care products, phenolics, and others (hydrocarbons, inorganic anions, etc.) [1]. The source of these noxious pollutants come from untreated discharges of irresponsible textile, pharmaceutical, agricultural and mining factories. Pharmaceutical pollutants have attracted interest among researchers to solve this problem which can cause serious health effects towards humans and may harm the environment. Research on the fate of pharmaceuticals in water prove that there are still traces found downstream [2]. Long-term exposure to low concentrations of pharmaceuticals in water and the synergistic effects of mixtures may pose risks to human health, especially to sensitive subpopulations [3–5]. It gains more concern as it is reported most pharmaceuticals are non-biodegradable [6–8]. Sulfamethoxazole (SMX) is regularly prescribed as an antibiotic either as individual medication or combined with other drugs such as trimethoprim, a combination commercially known as Bactrim. Bactrim is used to treat several diseases, for instance, many types of bacterial infection namely urinary tract infection including pneumonia. Known SMX contraindications cause allergy, hypersensitivity to either component, hyperkalaemia, severe hepatic or renal dysfunction [9]. Acetaminophen (ACM) are used as an analgesic for mild to moderate pain and an anti-pyretic [10]. The use of ACM is prevalent among surveyed consumers and has the highest consumption levels in Malaysia [11]. To put it more simply, because the water also contains other unwanted drugs, the incident of unconscious consummation of other drugs causes interaction with other drugs, thus leading to the worst condition in a prolonged period. It is known that not all drugs are completely digested in the human body which are then secreted as urine into wastewater and it is found that several amounts of concentration of SMX are found in water bodies; approximately 15% of the unmetabolized fraction goes into sewage after being ingested and subjected to human metabolism [12]. Table 1 shows the maximum concentration of SMX and ACM in surface water in Asia-Pacific and globally.

There are various removal methods that are used to remove pollutants from water such as bioreactors [14], degradation and photodegradation [15–18], filtration [19], photocatalysis [20,21], photo-Fenton and Fenton-like [22,23], coagulation [24] and adsorption [25,26]. Among these methods, adsorption is a widely used and most efficient method for wastewater treatment. This technique is superior and considered as powerful alternatives to conventional methods [27] because of its low cost, ease of operation, great efficiency, high capacity, simplicity, reliability and less energy consumption [28,29]. Graphene has gained interest as an adsorbent owing to its large surface area, accessible surface, high hydrophobicity and high mechanical properties [30,31]. As such, many researchers have explored graphene or graphite-based materials as an adsorbent for different pollutants [32–34] owing to its enormous, delocalized π-π electron system that can form strong bonding with various pollutants [35]. In previous research, graphene nanoplatelets (GNP) C750 have been studied as an adsorbent with good adsorption capacity, yet are quite expensive and other types of GNPs with different surface areas are unexplained [35,36]. In this study, the removal efficiency of SMX and ACM by the adsorption process from water using GNPs was evaluated. Different types of GNPs: C750, C300, M15 and M5, were screened to choose the best adsorbent and further optimized by several parameters: solution pH, GNP amount, initial concentration and contact time. Langmuir and Freundlich isotherms were applied to analyse the adsorption equilibrium. Pseudo-first order and pseudo-second order were also discussed for the kinetic study of removal capacity and to identify the behaviour and mechanism of adsorption. Molecular docking studies were also implemented to give interaction insight of GNP with SMX and ACM.

# 2. Materials and methods

## 2.1. Chemicals and materials

GNPs C750, C300, M15 and M5 were purchased from XG Sciences, USA. SMX, 98% was obtained from Alfa Aesar, USA and ACM (4-acetamidophenol), 98% purity was purchased from Acros Organic (table 2). Deionized water was used as a solvent for a stock solution and further dilution. The solution pH was measured using a HI-2002 Edge pH Meter, Hanna Instrument, USA.

**Table 1.** Maximum concentration of sulfamethoxazole and acetaminophen found in water [13].

| pharmaceutical substance | therapeutic group | Asia-Pacific ($\mu$g l$^{-1}$) | global ($\mu$g l$^{-1}$)[a] |
|---|---|---|---|
| sulfamethoxazole | antibiotics | 14.3 | 29.0 |
| acetaminophen | analgesics | 9.17 | 230.0 |

[a]Includes Asia-Pacific, Africa, Eastern Europe, Latin America and Caribbean, western Europe and others in United Nations regions.

**Table 2.** Physicochemical properties of pharmaceuticals.

| pharmaceutical | chemical formula | CAS No | solubility in water (mg l$^{-1}$) | dissociation constant, pKa | molecular weight (g mol$^{-1}$) |
|---|---|---|---|---|---|
| sulfamethoxazole | $C_{10}H_{11}N_3O_3S$ | 723-46-6 | 610 | 1.6 and 5.7 | 253.28 |
| acetaminophen | $C_8H_9NO_2$ | 103-90-2 | 14 000 | 9.5 | 151.16 |

## 2.2. Characterization of adsorbents

The compositional and morphological properties of GNPs were determined using a high-resolution transmission electron microscope (HRTEM) JEM-2100F operated at 200 kV (JEOL, Japan). Specimens were prepared by dispersing the GNPs in 100% acetone and adding a drop of dispersed GNP on the copper grid. Raman spectroscopy was performed using WITec Alpha 300R (WITec GmbH, Germany). Brunner–Emmett–Teller (BET) and Barrett–Joyner–Halenda (BJH) for specific surface area, pore volume, pore size and pore width distribution of GNPs were analysed using MicroActive for TriStar II Plus 2.03 (Micromeritics, USA), the samples were degassed with nitrogen gas at about 150°C with saturation pressure of approximately 756 mmHg overnight before analysis. X-ray diffraction analysis used LabX XRD-6000 (Shimadzu, Japan) by scanning the GNPs at $2\theta$ of 20°–80° at a speed angle of 2° min$^{-1}$ with Cu K$\alpha$ radiation. Zeta potential and average diameter were measured using Zetasizer Nano (Malvern Panalytical Ltd, UK).

## 2.3. Preparation of solution

The stock solution of SMX was freshly prepared by dissolving 10 mg in a 100 ml volumetric flask to produce 100 mg l$^{-1}$ of the pharmaceutical solution then further diluted to lower concentration using deionized water. The calibration curve of SMX was plotted by absorbance at 264 nm and ACM at 243 nm. The regression equation from EXCEL was used to calculate the concentration after the adsorption process, $C_e$.

## 2.4. Preliminary adsorption study

Different types of GNPs with different surface areas were used towards the removal of SMX and ACM to select the best adsorbent among four types of GNPs: C750, C300, M15 and M5. The study was carried out by preparing a 10 ml of 20 mg l$^{-1}$ SMX solution and using 20 mg of GNPs as adsorbent. The solution was then sonicated for 20 min which was later centrifuged, and the supernatant was filtered by using a 0.22 $\mu$m cellulose acetate (CA) syringe filter. The concentration of the supernatant was obtained by detecting absorbance using a UV-Vis spectrophotometer (Shimadzu UV-1650PC).

## 2.5. Batch adsorption experiments

Batch adsorption experiments were performed by preparing 10 ml of the pharmaceutical solution to investigate the effect of adsorption parameters: solution pH, GNP amount, initial concentration and contact time. The stock solution was prepared then further diluted to set the initial concentration. The solution pH was adjusted using 0.1 M NaOH and 0.1 M HCl and measured using a HI-2002 Edge pH Meter (Hanna Instrument, USA). GNPs were added into the solution and were sonicated continuously for a certain time. Then, the solution was centrifuged to collect the supernatant and

**Table 3.** Linearized expressions for each adsorption isotherm and kinetic.

| isotherms and kinetics | expressions | linear expression | plot |
|---|---|---|---|
| Langmuir isotherm | $q_e = q_m \left[ \dfrac{K_L C_e}{1 + K_L C_e} \right]$ | $\dfrac{1}{q_e} = \left[ \dfrac{1}{K_{LII} q_m} \right] \left[ \dfrac{1}{C_e} \right] + \dfrac{1}{q_m}$ | $\dfrac{1}{q_e}$ Vs $\dfrac{1}{C_e}$ |
| Freundlich isotherm | $q_e = K_F C_e^{1/n}$ | $\ln q_e = \ln K_F + \dfrac{1}{n} \ln C_e$ | $\ln q_e$ Vs $\ln C_e$ |
| pseudo-first-order kinetic | $q_t = q_e^{(1 - e^{-k_1 t})}$ | $\ln(q_e - q_t) = -K_1 t + \ln q_e$ | $\ln(q_e - q_t)$ Vs $t$ |
| pseudo-second-order kinetic | $q_t = \dfrac{k_2 t q_e}{k_2 t + 1}$ | $\dfrac{t}{q_t} = \left[ \dfrac{1}{q_e} \right][t] + \dfrac{1}{K_2 q_e^2}$ | $\dfrac{t}{q_t}$ Vs $t$ |
| intraparticle diffusion | $q_t = K_{id} t^{(0.5)}$ | $q_t = K_{id} t^{(0.5)} + C$ | $q_t$ Vs $t^{(0.5)}$ |

was filtered through a 0.22 µm CA syringe filter, to ensure there is no trace of GNP in the solution before running the sample through the UV-Vis spectrophotometer to avoid ambiguous results. The removal efficiency of the solution was acquired from the calculation by using the following equation:

$$\text{removal efficiency (\%)} = \frac{(C_i - C_e)}{C_i} \times 100 \,, \tag{2.1}$$

where $C_i$ is the initial concentration of the pharmaceutical solution (mg l$^{-1}$) and $C_e$ is the equilibrium concentration of the solution retrieved after the removal process (mg l$^{-1}$). The adsorption capacity was calculated according to the equation:

$$q_e = \frac{(C_i - C_e) \, V}{m} \,, \tag{2.2}$$

where $q_e$ is the amount of pharmaceutical adsorbed by the GNPs (mg g$^{-1}$) or known as adsorption capacity at equilibrium, $V$ is the volume of pharmaceutical solution (l) and $m$ is the mass of GNPs used (g). The reported values were the means of triplicates. Isotherms and kinetics studies were further studied to understand the adsorption mechanisms of SMX and ACM onto GNP C300. Table 3 exhibits the expressions of both isotherms and kinetics. For the adsorption isotherm, Langmuir type and Freundlich models were chosen while pseudo-first-order, pseudo-second-order and intraparticle diffusion model were applied to understand the kinetics.

## 2.6. Regeneration

A regeneration study was carried out using the optimized parameters for adsorption. The used adsorbent was dispersed in 10 ml of different eluents: 0.005 M NaOH, 0.005 M HCl and 5% ethanol-deionized water then sonicated for 30 min for desorption process. The washed adsorbent was then centrifuged, and the supernatant was discarded. The adsorbent was dried in the oven around 55°C. The dried adsorbent was reused for the adsorption process and the steps were repeated for several cycles.

## 2.7. Real water application

Three different real water samples: Langat river water, Kuyoh river water and laboratory tap water were collected for environmental application. The samples were filtered using filter paper and kept at 4°C in the dark. The properties of the three real water samples were analysed. Real water samples were spiked with a concentrated solution of ACM and SMX (100 mg l$^{-1}$) were added into the real water samples to obtain a sample with final concentrations of 20 mg l$^{-1}$. Twenty milligrams of GNPs was added to the spiked samples and sonicated for 20 min for the removal process. The sample was centrifuged, and the supernatant was filtered with a 0.22 µm CA syringe filter into a vial ready to be analysed. Chemical oxygen demand (COD, mg l$^{-1}$) and biochemical oxygen demand (BOD, mg l$^{-1}$) were measured following the APHA 5220 and APHA 5210 methods, respectively. Total soluble solids (TSS, mg l$^{-1}$) was analysed, a Hach DR 900 (Hach, USA) and a WTW Inolab meter (WTW GmbH,

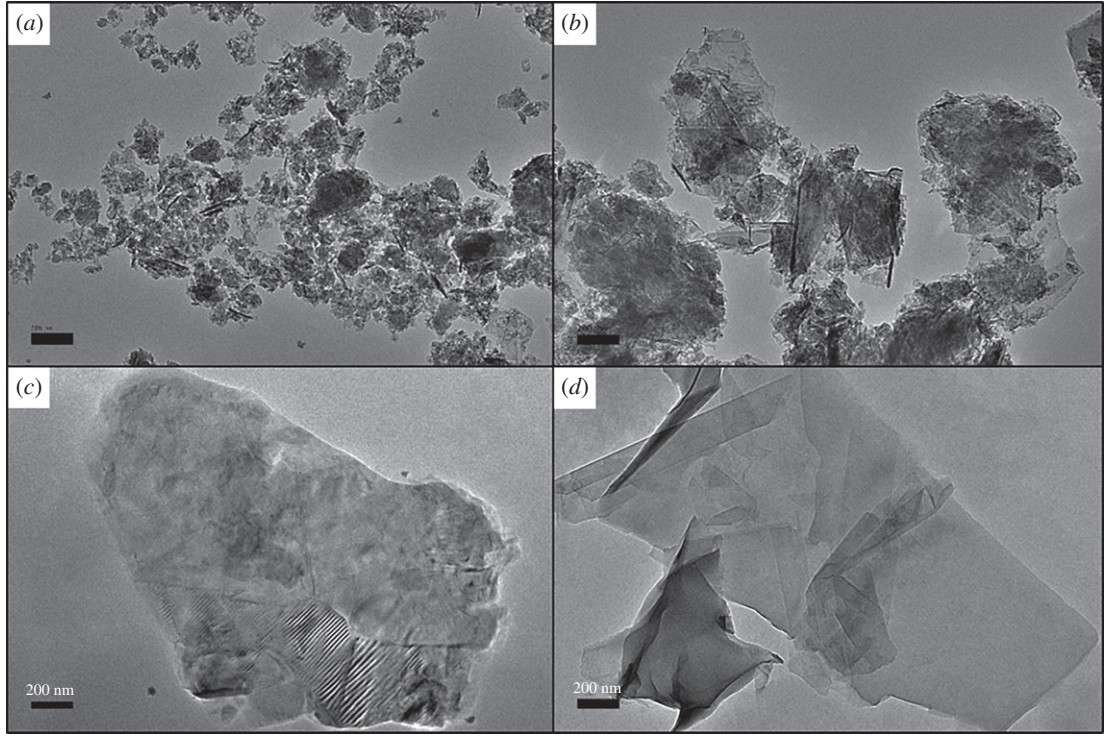

**Figure 1.** HRTEM images of (*a*) C750, (*b*) C300, (*c*) M15 and (*d*) M5 at 10 000 times magnification.

**Table 4.** Analysis of the Langat river water, Kuyoh river water and tap water.

|  | Langat river water | Kuyoh river water | tap water |
|---|---|---|---|
| chemical oxygen demand, COD (mg l$^{-1}$) | 43.63 | 77.57 | 14.60 |
| biochemical oxygen demand, BOD (mg l$^{-1}$) | 2.15 | 7.20 | 1.40 |
| total soluble solids, TSS (mg l$^{-1}$) | 35.00 | 18.00 | 0 |
| total dissolved solids, TDS (mg l$^{-1}$) | 171.00 | 282.00 | 150.00 |
| sodium, Na (mg l$^{-1}$) | 5.50 | 10.00 | 7.58 |
| magnesium, Mg (mg l$^{-1}$) | 1.25 | 1.84 | 1.27 |
| calcium, Ca (mg l$^{-1}$) | 13.32 | 24.86 | 14.64 |

Germany) was used to measure total dissolved solids (TDS, mg l$^{-1}$). The concentration of sodium, magnesium and calcium were analysed using inductively coupled plasma-optical emission spectrometry by referring to the APHA 31110b method (table 4).

# 3. Results and discussion

## 3.1. Characterization of graphene nanoplatelets

By using an HRTEM at 10 000 times magnification, physical features of GNPs were analysed. Figure 1 displayed the structure of GNPs which possessed wrinkle flakes with multiple layers. C750 (figure 1*a*) has an irregular shape which resembles more like platelets, more wrinkle features and smaller size with greater aggregation compared to C300, M15 and M5. As shown in figure 1*c,d*, M5 and M15 exhibit bigger flakes with a less overlapping layer.

The thickness of layer GNPs was evaluated at atomic resolution via Raman spectroscopy which shows vibrational bands of D, G and 2D (figure 2). The D band, known as the disorder band appears around 1350 cm$^{-1}$ because of out of plane vibrations attributed to the presence of structural defects. The presence of this band proved the GNPs have ragged edges [37,38]. This result is also confirmed in the

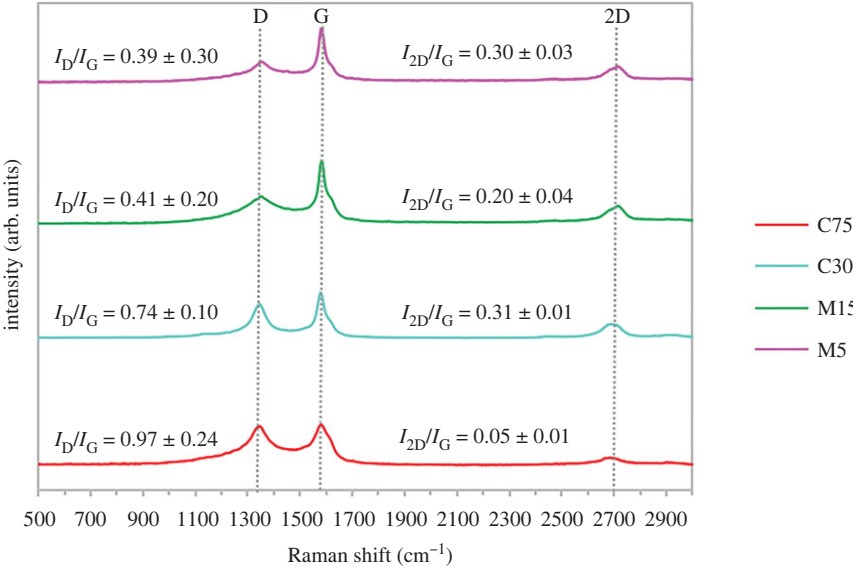

**Figure 2.** Raman spectra of GNPs C750, C300, M15 and M5.

HRTEM images. The G band which is a sharp peak that exists around 1587 cm$^{-1}$ resulted from the in-plane vibration of sp$^2$ hybridized carbon atoms. The 2D band is the second order of the D band, sometimes denoted as the G' band that appears around 2700 cm$^{-1}$, resulted from a two-phonon lattice vibrational process without defect. The shape of 2D bands was wide and low and the full width at half maximum of the 2D bands for all GNPs greater than 90 cm$^{-1}$ perfectly categorized as multilayer graphene [39]. Intensity ratio of 2D and G peaks were used to determine the number of graphene layers. The ratios $I_{2D}/I_G$ of four different GNPs are less than 1, thus confirming the multilayer properties of these adsorbents.

Figure 3 shows the X-ray diffraction patterns of GNP C750, C300, M15 and M5. Peaks shown in the figure correspond with the characteristic peaks of pristine graphite which could be assigned to the (002), (100), (004) and (110) hexagonal planes by referring to the JCPDS Card No (75-1621). C300 (figure 3b) shows more distinct peaks with a higher intensity that resemble graphite compared with the X-ray diffraction pattern of C750 (figure 3a) that only exhibits peaks of (002) and (100) with a broader peak.

Through BET and BJH analysis, the surface area, micropore area, pore size and pore volume of adsorbent were analysed and summarized in table 5. The surface area C750 (738.44 m$^2$ g$^{-1}$) was found to be the highest compared to others followed by C300 (390.30 m$^2$ g$^{-1}$), M15 (61.51 m$^2$ g$^{-1}$) and M5 (63.55 m$^2$ g$^{-1}$). The pore diameter of C750, C300, M15 and M5 can be categorized as mesopores as they fall in the range of (2–50 nm) according to the International Union of Pure and Applied Chemistry classification.

Figure 4 exhibits adsorption/desorption isotherm type IV which is similar to reported by Al-Khateeb *et al.* [36] with an H3 type hysteresis loop that confirms there were aggregates (loose assemblages) of platelet like particles forming slit-shaped mesopores [40]. The intermediate flat region in the isotherm corresponds to monolayer formation at the lower pressure region of the graph then followed by multilayer [41].

Zeta potential measured the electrostatic repulsion between similarly charged particles and hence indicates a degree of dispersion efficiency thus giving information tendency of agglomeration, and a larger zeta potential may mean there is a better dispersion [42,43]. Table 6 exhibits the measured zeta potential of GNP C300 (−50.80 mV) is the most stable compared to other three GNPs. GNP M5 has the least absolute value with −20.87 mV, which considered its tendency to aggregate since it is lower than −30 mV [44]. Both GNP C750 and C300 possessed an average diameter of less than 2 μm, while M5 and M15 with 4.21 μm and 6.79 μm, respectively.

## 3.2. Preliminary adsorption study

Figure 5 presents the preliminary adsorption study of four different types of GNPs C750, C300, M15 and M5 towards SMX. Based on the characterization, C750 possessed the advantage of a higher surface area of the four materials tested. However, it is found that C300 has the highest removal efficiency compared to other types of GNPs despite the fact C750 has a higher surface area. Furthermore, C300 has the highest stability based on the zeta potential. This finding was contrary to expectations that a higher surface area will have

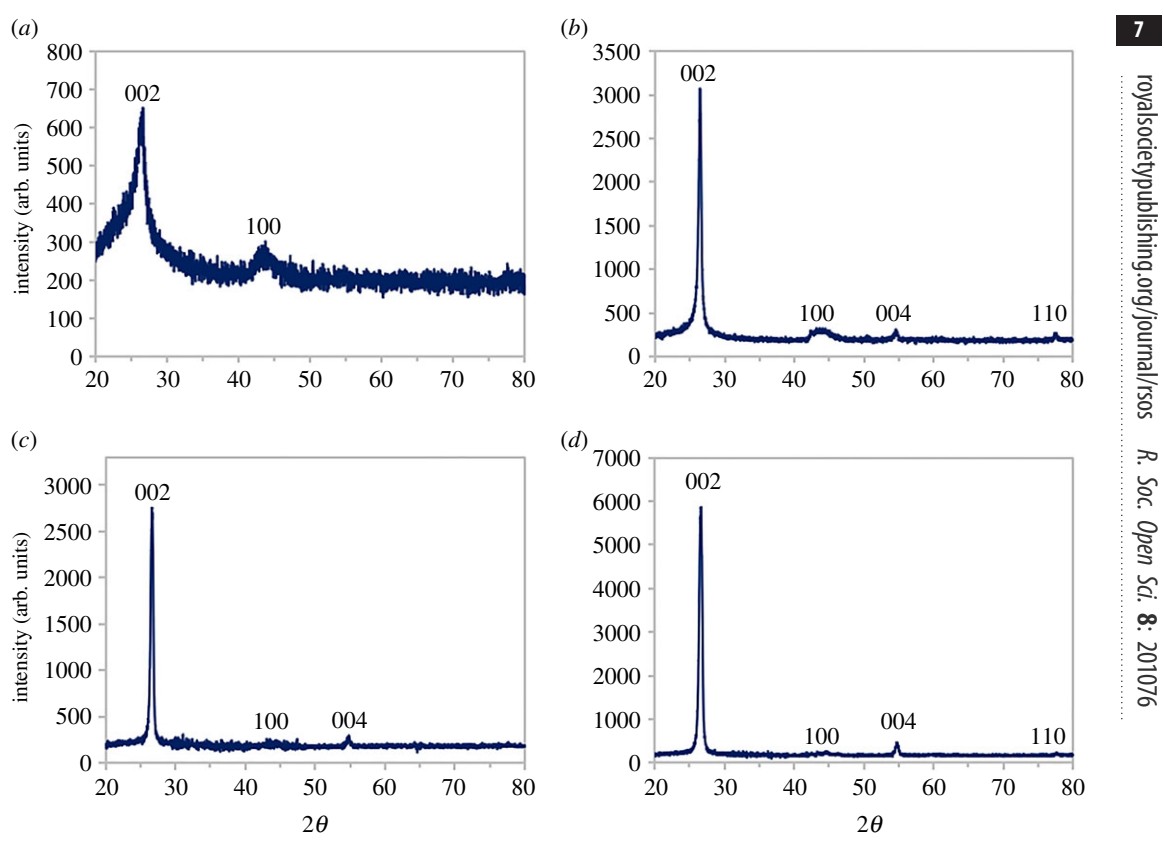

**Figure 3.** X-ray diffraction spectra for GNPs (*a*) C750, (*b*) C300, (*c*) M15 and (*d*) M5.

**Table 5.** Summary of BET and BJH analysis for GNP C750, C300, M15 and M5.

| GNP | BET surface area ($m^2 \, g^{-1}$) | micropore area ($m^2 \, g^{-1}$) | total pore volume ($cm^3 \, g^{-1}$) | external surface area ($m^2 \, g^{-1}$) | BJH pore width (nm) |
|------|--------|--------|------|--------|------|
| C750 | 738.44 | 218.71 | 0.78 | 519.73 | 4.96 |
| C300 | 390.30 | 83.65 | 0.41 | 306.65 | 4.66 |
| M15 | 61.51 | 10.00 | 0.10 | 51.51 | 5.38 |
| M5 | 63.55 | 17.96 | 0.08 | 45.59 | 4.94 |

better adsorption efficiency [45], though the result of removal efficiency is similar to what has been reported by Al-Khateeb [35]. Thus, other factors must be taken into consideration to achieve a higher percentage of adsorption. This might be owing to clumping or aggregation of C750 because of its high surface area which causes lower yet good adsorption of SMX. After centrifugation and filtration, some GNPs may have not filtered thoroughly owing to the small size of C750 able to go through 0.22 µm CA syringe filter. Because the objective of this preliminary study was to choose the best adsorbent, C300 was chosen. On top of that, C300 has a lower manufacturing cost compared to C750 which reduces the total cost for the overall process [46,47]. Thus, C300 was further used for the batch adsorption study with different parameters: solution pH, GNP amount, initial concentration of SMX and contact time.

## 3.3. Evaluation of adsorption behaviours of graphene nanoplatelets C300 towards sulfamethoxazole and acetaminophen

### 3.3.1. Effect of solution pH

To assess the adsorption behaviours of GNP C300, a batch adsorption study was performed to remove SMX and ACM by using GNP C300 as adsorbent. Solution pH is an important factor in the adsorption of

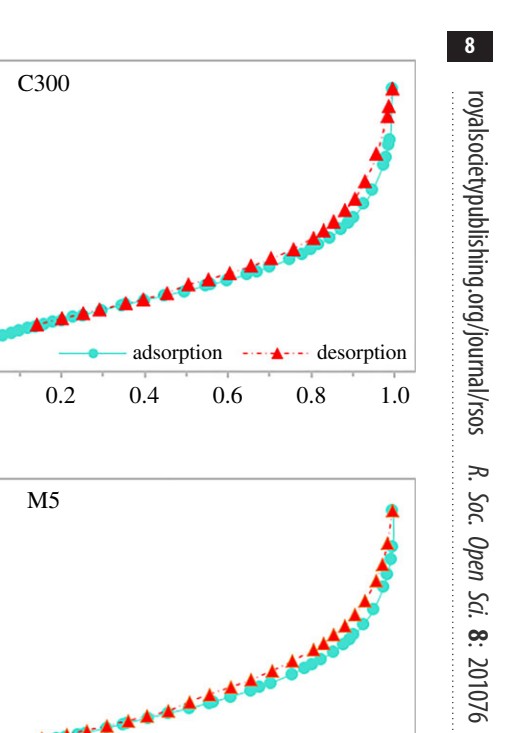

**Figure 4.** $N_2$ adsorption/desorption isotherms for GNP (*a*) C750, (*b*) C300, (*c*) M15 and (*d*) M5.

**Table 6.** Zeta potential and average diameter of different types of graphene nanoplatelets using dynamic light scattering.

| GNP | average diameter (nm) | zeta potential (mV) |
|-----|------------------------|---------------------|
| C750 | 933.67 | −44.63 |
| C300 | 1010.00 | −50.80 |
| M15 | 6794.67 | −33.90 |
| M5 | 4205.33 | −20.87 |

pharmaceuticals. The effect of solution pH on the adsorption of pharmaceuticals was studied from pH 2 to 10 in a 10 ml solution volume of $20 \, \mathrm{mg \, l^{-1}}$ concentration using 20 mg GNP C300 for 20 min. As displayed in figure 6, the adsorption ability of GNP C300 towards both SMX and ACM was affected by the pH value of the solution. As the initial pH of the solution increased from 2 to 4, the adsorption capacity of SMX onto GNP C300 gradually increased (figure 6*a*). On the other hand, the adsorption capacity and removal efficiency for ACM increased gradually from acid to neutral then decreased as it further increased (figure 6*b*). Maximum adsorption was detected at pH 4 (99.27%) for SMX and ACM at pH 8 (98.38%). The lowest removal for SMX was detected at pH 10 with 85.65% and for ACM at 76.8% at pH 2. The increasing solution pH was unfavourable for the removal of SMX molecules and resulted in a low adsorption capacity at higher pH. Because SMX is known as an acidic organic compound with dissociation constant (pKa) of 1.6 and 5.7, the removal of the targeted compound was better under acidic conditions ($pH < pKa$) than at basic pH ($pH > pKa$). In other words, when solution pH is less than their pKa values, they are present in aqueous solution as a protonated form and these observations are in agreement with the results in the previous works [33,48].

### 3.3.2. Effect of graphene nanoplatelets amounts

Different amounts of GNP C300 were used for adsorption towards a 10 ml of $20 \, \mathrm{mg \, l^{-1}}$ adjusted to pH 4 in 20 min for SMX and pH 8 for ACM based on previous optimum conditions. From the results presented in

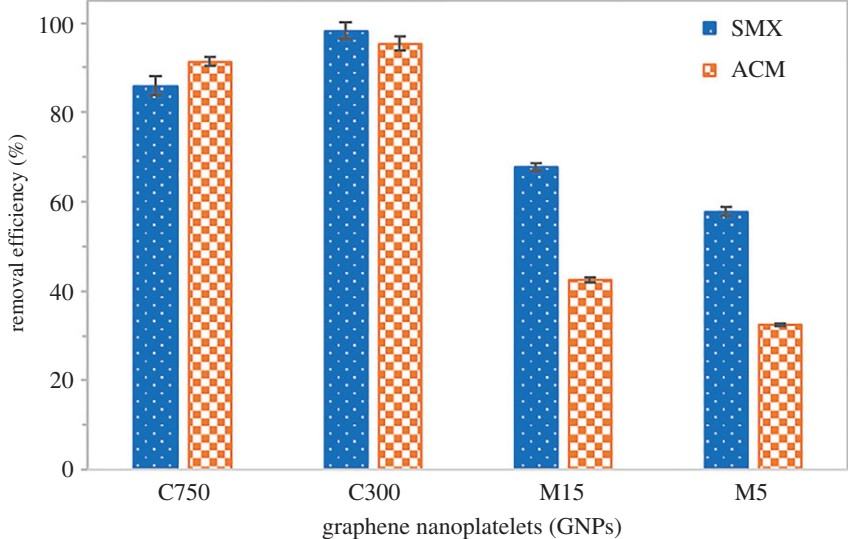

**Figure 5.** Preliminary test of different type of graphene nanoplatelets (GNPs).

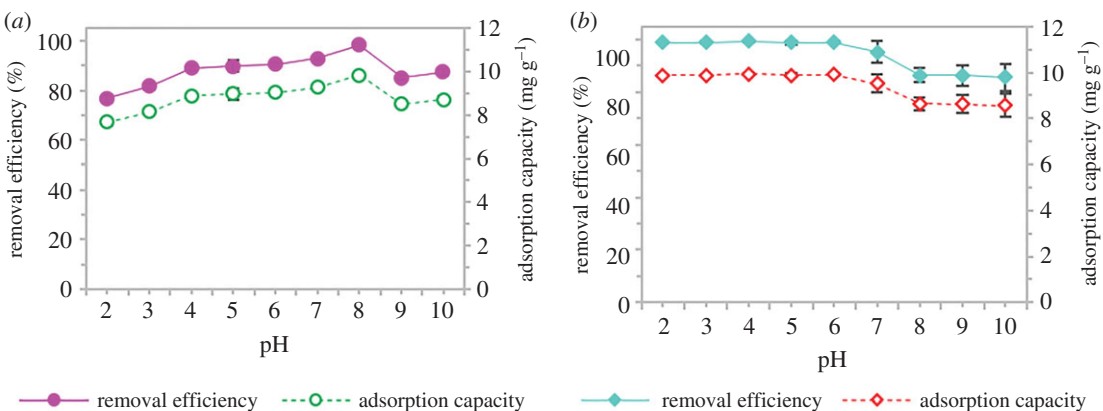

**Figure 6.** The effect of solution pH on (*a*) sulfamethoxazole and (*b*) acetaminophen using adsorption by GNP C300.

figure 7, with the increase in GNP C300 amount from 5 mg to 60 mg, the removal efficiency increased; however, the adsorption capacity for both pharmaceuticals decreased. At low adsorbent dosage, adsorption sites were fully available, resulting in higher adsorption capacity. At a higher GNP C300 amount, the active sites of the adsorbent which prepared to bind with the pharmaceutical molecules were not fully used in comparison to lower adsorbent dosages. The increase in the amount of GNP C300 may cause aggregation of adsorbent and possibility of collisions between solid particles [49,50], and consequently, the available adsorption sites may decrease as well owing to the adsorption capacity of GNP C300 for SMX and ACM.

### 3.3.3. Effect of initial concentration

It is crucial to study the effect of initial concentration upon revealing any adsorption process. Thus, a set of experiments was carried out at different initial concentrations of 10 to 100 mg l$^{-1}$ using previously optimized parameters. Figure 8 shows that the adsorption capacity of pharmaceutical increased as the initial SMX and ACM concentration increased from 10 to 100 mg l$^{-1}$ with an almost linear trend. The higher the initial concentration, the more adsorbate molecules available for adsorption, resulting in a higher adsorption capacity at higher pharmaceutical concentrations. However, as the initial concentration increased more than 40 mg l$^{-1}$, the removal efficiency slightly decreased. Nevertheless, the range of removal efficiency is still more than 98%.

### 3.3.4. Effect of contact time

The contact time between both pharmaceuticals and C300 was studied with several sets of batch adsorption experiments using the previously optimized condition at different times of 2 to 60 min,

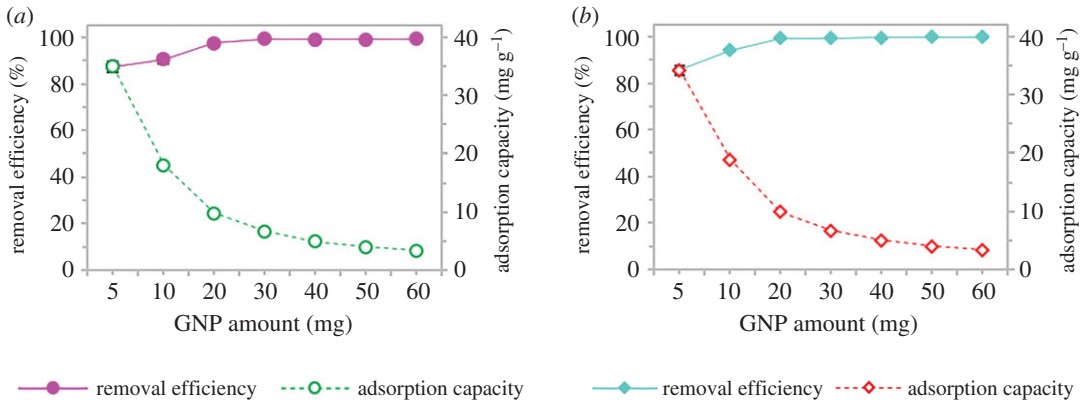

**Figure 7.** The effect of GNP amount on (*a*) sulfamethoxazole and (*b*) acetaminophen using adsorption by GNP C300.

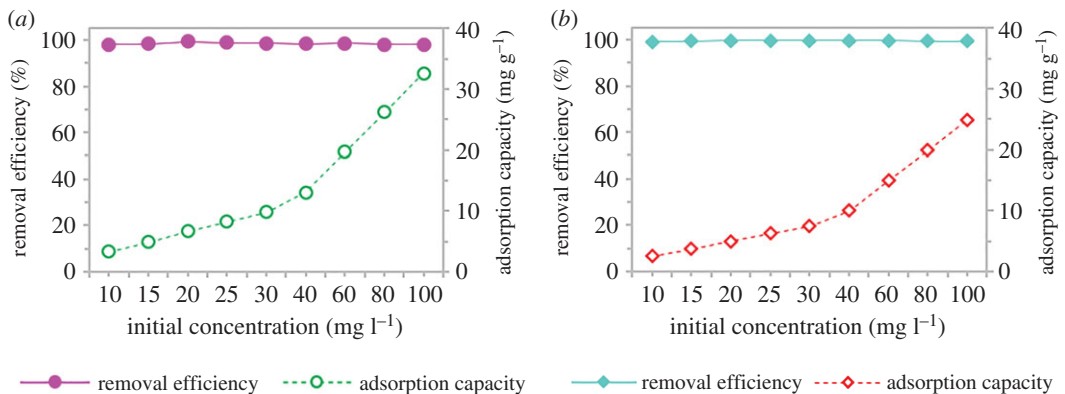

**Figure 8.** The effect of initial concentration on (*a*) sulfamethoxazole and (*b*) acetaminophen using adsorption by GNP C300.

and the results are presented in figure 9. As shown in the figure, SMX and ACM adsorption by GNP C300 occurred very fast initially and then the rate slowed down as the contact time increased. Both adsorption capacity and removal efficiency have the same trend. During the initial period, the adsorption capacity of the pharmaceuticals rapidly increased with time as many active sites were available on the surface of GNP C300 for binding of pharmaceutical molecules [51]. These available active sites were gradually occupied as time increased thus resulting in reducing adsorption rate [50].

### 3.3.5. Adsorption isotherm and kinetic

Langmuir and Freundlich isotherm models were applied to investigate the adsorption equilibrium between the pharmaceuticals and GNP C300. As shown in table 7, the adsorption data fitted well with both Langmuir and Freundlich isotherms and this finding was similarly reported in [52]. The $R^2$ value obtained from the Langmuir isotherm model were 0.9952 for SMX and 0.9946 for ACM, revealing that the Langmuir isotherm model slightly fit better and is more suitable for describing the adsorption process of the pharmaceuticals onto GNP C300. This illustrated that adsorption of SMX and ACM on GNP C300 was monolayer adsorption occurring on a homogenous surface without interaction between the adsorbate.

Adsorption kinetic with a sample concentration of $20 \, \text{mg} \, \text{l}^{-1}$ was further studied, and the kinetic parameters as well as the correlation coefficient, $R^2$ were acquired by linear regression. According to the data presented in the table, the pseudo-second-order model fitted better than the pseudo-first-order kinetic models. Based on the pseudo-second-order model, the correlation coefficients ($R^2$) value was 0.9999 for both SMX and ACM which was greater than pseudo-first order. Thus, the pseudo-second-order model was more suitable for describing the process of SMX and ACM adsorption on GNP C300 in agreement with chemisorption being the rate-controlling step [53–55]. The intraparticle diffusion model by Weber and Morris derived from Fick's second law of diffusion was applied to

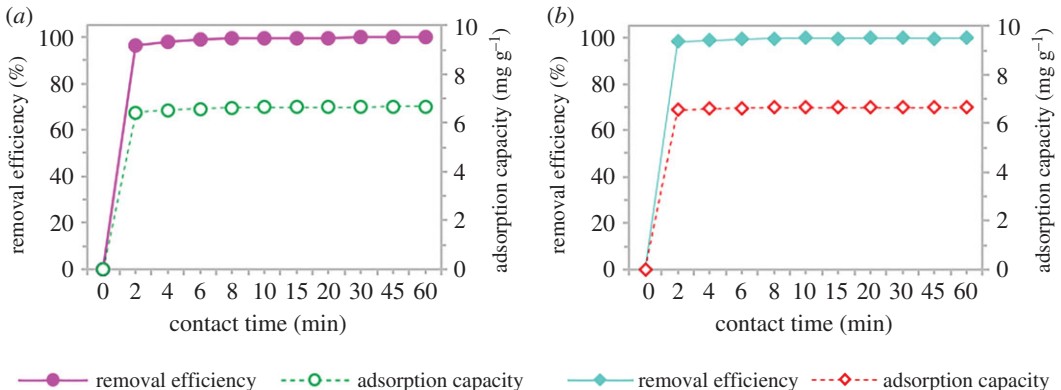

**Figure 9.** The effect of contact time on (*a*) sulfamethoxazole and (*b*) acetaminophen using adsorption by GNP C300.

**Table 7.** Comparison of adsorption isotherm and kinetic for sulfamethoxazole and acetaminophen.

| isotherm | | | | | | |
|---|---|---|---|---|---|---|
| | Langmuir | | | Freundlich | | |
| | $n$ | $K_{L2}$ | $R^2$ | $n$ | $K_F$ | $R^2$ |
| SMX | 210.084 | 0.374 | 0.9952 | 1.193 | 52.034 | 0.9848 |
| ACM | 56.211 | 0.721 | 0.9946 | 1.435 | 21.230 | 0.9842 |
| kinetic | | | | | | |
| | pseudo-first order | | | pseudo-second order | | |
| | $K_1$ | $Q_e$ | $R^2$ | $K_2$ | $Q_e$ | $R^2$ |
| SMX | 0.108 | 0.129 | 0.6743 | 7.059 | 6.653 | 0.9999 |
| ACM | 0.115 | 0.435 | 0.8218 | 1.815 | 6.667 | 0.9999 |

explain further on the diffusion mechanism which includes three steps. The first two steps are transport steps, and the last step is a reaction step [1]: (i) film diffusion (external diffusion), which is the transport of adsorbate from the bulk phase to the external surface of the adsorbent; (ii) pore diffusion, which is the transport of adsorbate from the external surface into the pores; and (iii) surface reaction, which the adsorbate will attach to the internal surface of the adsorbent. The data show multilinearity (electronic supplementary material, table S1) which indicates that the intraparticle diffusion controls the adsorption process within the early stage, but was not the rate-determining step for the whole adsorption process similarly found in previous research [35].

The maximum adsorption capacities for both ACM and SMX from this paper were compared with other adsorption capacities reported in the literature. Table 8 summarizes the previous research using graphene and activated carbon as adsorbent using the adsorption method.

## 3.4. After adsorption analysis

Table 9 compares the BET results before and after the adsorption of SMX and ACM. After the adsorption process, one can see the surface area of adsorbent decreased to that before the adsorption process, C300 (390.30 m$^2$ g$^{-1}$) while after the adsorption process, C300-SMX (325.62 m$^2$ g$^{-1}$) and C300-ACM (291.70 m$^2$ g$^{-1}$). This shows adsorption had occurred as the SMX molecules adhered to the surface of GNP C300 thus resulting in a notably lower surface area, micropore area and external surface area.

Field emission scanning electron microscopy (FESEM)/energy dispersive X-ray (EDX) was used to observe the adsorbent surface and elemental analysis (figure 10). After the adsorption process, other

**Table 8.** Previous research on sulfamethoxazole and acetaminophen.

| adsorbents | pollutant | adsorption capacity (mg g$^{-1}$) | references |
|---|---|---|---|
| graphene nanoplatelet C300 | SMX | 210.08 | this work |
| powder activated carbon | SMX | 86 | [56] |
| mesoporous silica-magnetic graphene oxide nanocomposite | SMX | 15.46 | [34] |
| multi-walled carbon nanotubes | SMX | 29 | [57] |
| graphene nanoplatelet C300 | ACM | 56.21 | this work |
| graphene nanoplatelets C750 | ACM | 18.07 | [35] |
| graphene | ACM | 18.28 | [58] |
| activated carbon from *Quercus Brantii* (oak) | ACM | 45.45 | [59] |

**Table 9.** Comparison before and after adsorption of SMX and ACM towards GNP C300 using BET analysis.

| | BET surface area (m$^2$ g$^{-1}$) | micropore area (m$^2$ g$^{-1}$) | total pore volume (cm$^3$ g$^{-1}$) | external surface area (m$^2$ g$^{-1}$) |
|---|---|---|---|---|
| C300 | 390.30 | 83.65 | 0.41 | 306.65 |
| C300-SMX | 325.62 | 38.33 | 0.37 | 287.29 |
| C300-ACM | 291.70 | 11.86 | 0.36 | 279.84 |

elements were present on the surface of GNP C300 of other elements besides carbon. The elements of oxygen and nitrogen were detected on the surface of GNP C300 after adsorption of SMX and ACM. There was also the presence of sulfur found on the C300-SMX sample.

To get a better view of the adsorption mechanisms, molecular docking was applied, and the data are detailed in table 10. Binding energy measures the affinity of the GNP-SMX complex and the sum of energies of each molecule separately. The lower the energy value, the more stable it is. The binding energies for both molecules were negative values; −7.54 kcal mol$^{-1}$ and −5.29 kcal mol$^{-1}$ for SMX and ACM, respectively, which means the SMX and ACM were bound spontaneously to the surface of GNP C300 without consuming energy. The ΔGbind obtained for GNP-SMX is higher 2.25 kcal mol$^{-1}$ than GNP-ACM which is in good agreement with experimental. In the experimental results, GNP shows high maximum adsorption capacity towards SMX (210.08 mg g$^{-1}$) compared to ACM (56.21 mg g$^{-1}$). This different adsorption capacity of GNP towards pollutants can be explained by the strength of the binding energy obtained from the docking results. Both pollutants preferred to bind at the surface layer of GNPs. Van der Waals, hydrogen bonds and desolvation energy described energy loss with the interaction between pharmaceuticals and GNP and solvent upon binding which revealed the chemisorption occurred exothermically (electronic supplementary material, figure S2).

The adsorption mechanism of GNP towards the targeted pollutants can take place in the form of electrostatic interactions (cation or anion attraction), π–π interaction, hydrophobic interaction and hydrogen bonding [60,61]. Both SMX and ACM molecules contain an aromatic group or benzene ring (electronic supplementary material, figure S3). The adsorption mechanism is mainly owing to interaction of the electron-rich GNP surface and the protonated aniline ring of pharmaceutical molecules. When the pH is near or equal to the pKa of pharmaceutical, the pharmaceutical molecules exist in neutral form thus it favours hydrogen bonding and π–π interaction with GNPs. When the solution pH drifted to more acid or basic conditions, electrostatic interactions will occur between the GNP and pharmaceutical molecules.

## 3.5. Regeneration

As depicted in figure 11, the removal efficiency for both SMX and ACM using GNP C300 through four regeneration cycles by different eluents: 0.005 M NaOH, 0.005 M HCl and 5% ethanol-deionized water is shown. By using chemical regeneration, interaction forces formed between the

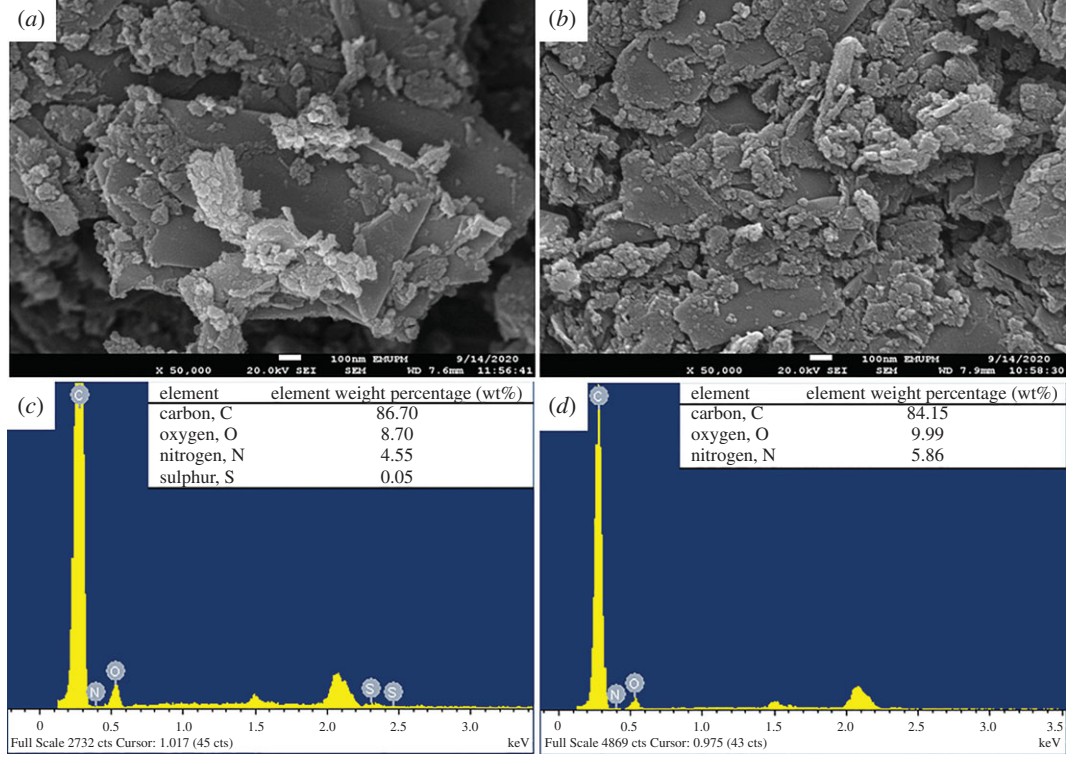

**Figure 10.** FESEM images of GNP C300 after adsorption of (a) SMX and (b) ACM. EDX analysis of GNP C300 after adsorption of (c) SMX and (d) ACM.

**Table 10.** Molecular docking of GNP-SMX and GNP-ACM. (VDW, Van der Waals.)

|  | binding energy, $\Delta G$bind (kcal mol$^{-1}$) | VDW + H-bond + desolvation energy (kcal mol$^{-1}$) | electrostatic energy (kcal mol$^{-1}$) |
|---|---|---|---|
| GNP-SMX | −7.54 | −8.74 | 0.00 |
| GNP-ACM | −5.29 | −5.89 | 0.00 |

adsorbed pharmaceutical pollutants and the eluent molecules causing the SMX and ACM molecules to leave the adsorption sites of the adsorbent and dissolve in the eluent [62]. Among the used eluents for the adsorption of SMX, 0.005 M HCl and 5% ethanol-deionized water were able to regenerate GNP C300 where high removal efficiencies could be achieved even after the fourth cycle with 98.79%–95.18% and 98.33%–95.95%, respectively. When washed with 0.005 M NaOH, graphene efficiency was reduced from 99.28% to 78.8% in the second cycle, having the lowest percentage after the fourth cycle (32.84%). GNP C300 was able to remove 95.58%–71.45% of ACM when washed with 5% ethanol-deionized water which is the highest followed by 0.005 M NaOH and 0.005 M HCl, with removal efficiencies of 96.01%–64.35% and 97.23%–26.08%, respectively. The surface of GNP was held by weak intermolecular interactions viz. Van der Waals forces and dipole–dipole interaction [38]. After several cycles, the removal efficiency of GNP is reduced because of the loss of GNP during the washing process as it was not fully separated by centrifugation. Conclusively, regeneration of the GNP C300 by 5% ethanol-deionized water was considered the suitable eluent for both pollutants, SMX and ACM with high removal efficiencies after several adsorption–desorption cycles.

## 3.6. Real water application

Based on figure 12, it can be concluded that GNP C300 managed to remove most of the pharmaceutical pollutants from the spiked real water samples. More than 98% of SMX were removed from all three real

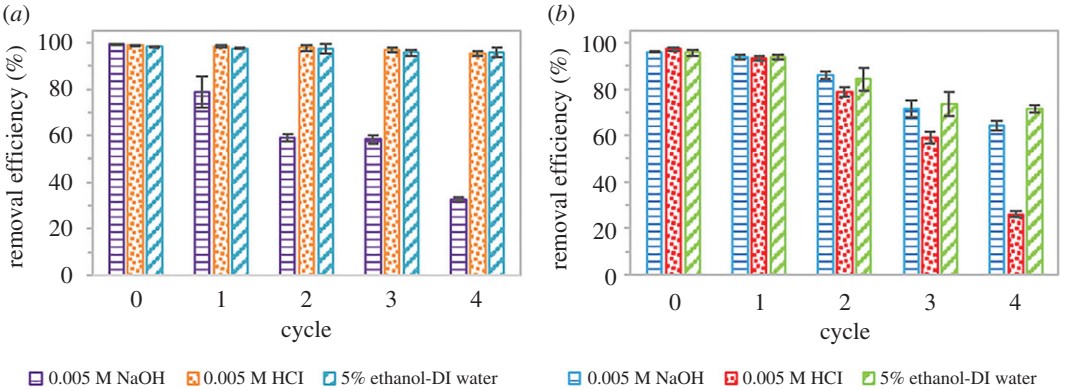

**Figure 11.** Removal efficiency of GNP C300 (*a*) SMX (*b*) ACM versus regeneration cycle using different eluents.

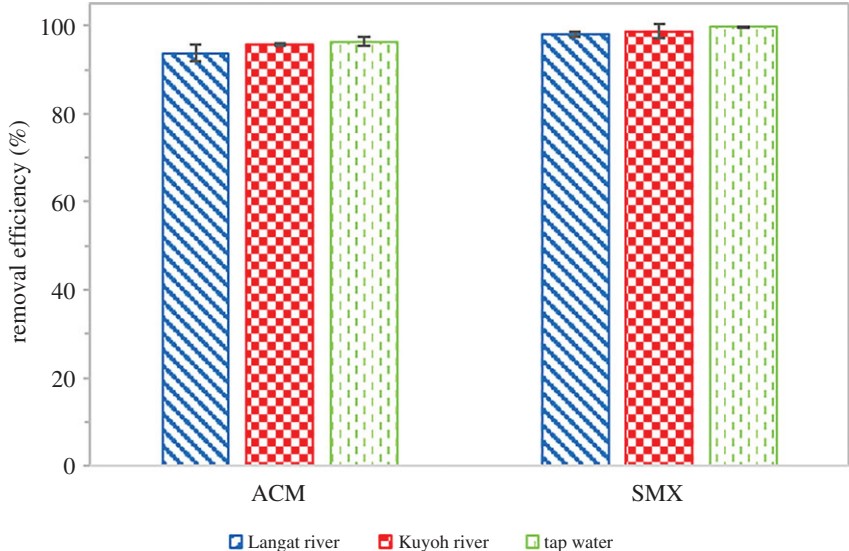

**Figure 12.** Removal efficiency of GNP C300 on real water samples; Langat river water, Kuyoh river water and tap water.

water samples: Langat river water, Kuyoh river water and tap water, while for ACM the removal efficiency was somewhat lower than SMX yet more than 90%. As shown in the figure, GNP C300 able to remove, respectively, 93.83%, 95.76% and 96.45% of ACM, as for SMX 98.10%, 98.79% and 99.88% from Langat river water, Kuyoh river water and tap water. The results proved that GNP C300 is also suitable for the real water application.

# 4. Conclusion

In conclusion, all four types of GNPs were able to remove SMX and ACM from water. GNP C300 shows the highest adsorption capacity compared to GNP C750, M15 and M5. The unexplored potential of GNP C300 and the rest has revealed the other types GNP that work better compared to C750, which is a bit costly and faces an agglomeration problem. The removal of pharmaceuticals with different pKa values was strongly affected by pH. It is proved that the adsorption of SMX and ACM using GNP C300 was owing to chemisorption as the data were best fitted with pseudo-second order. The adsorption data fitted well with both Langmuir and Freundlich isotherms. Owing to slightly higher $R^2$, the Langmuir model was more suitable for describing the adsorption process of GNP C300 towards both pharmaceuticals. In addition, GNP C300 can remove up to 99% of SMX and ACM. On top of that, a negative value of binding energy from docking showed that the adsorption was spontaneous and exothermic. In addition, 5% ethanol-deionized water was found to be the best eluent for the regeneration of GNP C300 as adsorbent for the removal of SMX and ACM. Real water application

results showed that GNP C300 managed to remove most of the pharmaceutical pollutants from the environmental samples. GNP C300 exhibits great potential as an excellent adsorbent with high adsorption ability for the removal of SMX and ACM, and thus may also be used towards other pharmaceutical pollutants.

Data accessibility. Data is available from the Dryad Digital Repository: https://doi.org/10.5061/dryad.qrfj6q5cv [63].

Authors' contributions. F.A.R. contributed to the laboratory works, acquisition of data, analysis and interpretation of data, drafting the article and revising it critically. K.J. and N.A.F.A. contributed to analysis specifically on the computational data of molecular docking. S.K., H.A. and A.H.A. reviewed and commented drafts of the manuscript. All authors gave approval for publication.

Competing interests. There are no competing interests to declare.

Funding. This project was financially supported by Universiti Putra Malaysia under High Impact Putra grant no. (UPM/700-2/1/GBP/2018/9598600).

Acknowledgements. The authors gratefully acknowledge Department of Chemistry, Faculty of Science, Universiti Putra Malaysia for supporting this work. We would also like to thank Martin R. Gill for his contribution in reviewing the manuscript.

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
