## [Reviewer comments · Royal Society Open Science]

Review History

RSOS-201076.R0 (Original submission)

Review form: Reviewer 1

Is the manuscript scientifically sound in its present form?

Yes

Are the interpretations and conclusions justified by the results?

No

Is the language acceptable?

Yes

Do you have any ethical concerns with this paper?

No

Have you any concerns about statistical analyses in this paper?

No

Recommendation?

Major revision is needed (please make suggestions in comments)

Comments to the Author(s)

In this study, graphene nanoplatelets (GNPs) were used to efficiently remove antibiotics sulfamethoxazole (SMX) and analgesic acetaminophen (ACM) as pharmaceutical pollutants from water by adsorption process. Different types of GNPs; C750, C300, M15, and M5 were characterized by high-resolution transmission electron microscopy (HRTEM), Raman spectroscopy, X-Ray diffraction (XRD) and BrunauerEmmett-Teller (BET). The effects of several parameters; viz., solution pH, adsorbent amount, initial concentration and contact time were studied. The parameters were optimized by batch adsorption process and the maximum removal efficiency for both pharmaceuticals were 99%. The adsorption kinetics and isotherms were studied, and the experimental data were best analysed with pseudo-second kinetic and Langmuir isotherm model with maximum adsorption capacity (Q_m) of 210.084 mg g⁻¹ for SMX and 56.211 mg g⁻¹ for ACM. The work is interesting and well documented but needs substantial revision before the acceptance:

1. Please describe the theoretic reasons on the adsorption of SMX and ACM using graphene nanoplatelets (GNPs).
2. How about the preparation reproducibility of the graphene nanoplatelets (GNPs).
3. The characterization of any adsorbent is very important to know the surface morphology, pore size, surface area and functional groups after adsorption too. The authors must include SEM, FTIR, and BET after adsorption of SMX and ACM also and the changes after the adsorption should be explained clearly.
4. Adsorption desorption and regeneration studies must be performed for the recyclic use of the adsorbent using different eluents.
5. An English review is required in this manuscript. There are several misspelling words throughout the text. Punctuation through the text is weak, and also should be checked.
6. Why the authors focused SMX and ACM for the removal study. Did they check any adsorption efficiency towards SMX and ACM.

There are many papers published on this topic and this one follows the same structure and content as many of the previously published ones, without adding much of significance to the current knowledge. What is the novelty in this work. How the adsorption technique is better than other methods. There are some other pollutants also exist with pharmaceutical pollutants like metals and dyes which may be discussed here. The merits and demerits of adsorption technique must be added in this part. The introduction part must be improved with latest references for the superiority of adsorption for the removal of different types of pollutants. As a suggestion, following references or other similar work may be followed: Journal of molecular liquids 260, 342-350, 2018; Chemical Engineering Journal 334, 462-478, 2018; Journal of Cleaner Production 250, 119553, 2020

9. Photodegrataion is also one of the commonly use methods?? As a suggestion, the authors can follow the following work or other similar work for the discussion: Desalin Water Treat 102, 273-279, 2018; Journal of Water Process Engineering 37, 101325, 2020; Journal of hazardous materials 364, 429-440, 2019

9. Include the breakthrough studies also. Did the authors check the zeta potential also?

10. What is the practical application of the work.

11. Quality of all figures must be improved to make them clear. Improve the resolution as well as size of the figs.

Review form: Reviewer 2 (Ali Zeraatkar Moghaddam)

Is the manuscript scientifically sound in its present form?

Yes

Are the interpretations and conclusions justified by the results?

No

Is the language acceptable?

Yes

Do you have any ethical concerns with this paper?

No

Have you any concerns about statistical analyses in this paper?

No

Recommendation?

Major revision is needed (please make suggestions in comments)

Comments to the Author(s)

Dear Authors,

The manuscript titled with "Efficient Removal of Pharmaceuticals from Water Using Graphene Nanoplatelets as Adsorbent" appears to provide a feasible adsorbent for efficient removal of pharmaceuticals from water samples. The manuscript is an interesting work and well presented, and I find this manuscript is suitable for publication in after major revision. The comments are listed below:

- 1) Please do not use the abbreviations in the abstract section.
- 2) The characterization section has detailed discussion, but FTIR analyses are suggested.
- 3) The intra-particle diffusion model study is recommended in kinetic study section.
- 4) The thermodynamic study is highly recommended.
- 5) As can be seen in many similar works which introduced new adsorbents, the desorption and reusability tests is necessary.
- 6) To investigate the applicability of the proposed new adsorbent for pollutant removal, the dye adsorption process from real samples, e.g. related industrial wastewaters, are recommended.
- 7) The discussion about the dye adsorption mechanism is not convincing. A coherent mechanism study must be done especially about the possible interactions between adsorbent and pharmaceuticals.
- 8) The authors should more explain about advantages of the proposed adsorbent and compared with the other similar adsorbent.
- 9) More explain about novelty of the work must added in the conclusion section.

Yours sincerely

Ali Zeraatkar Moghaddam,

Assistant Professor of Analytical Chemistry,

University of Birjand, Birjand, South Khorasan, Iran

E-mail address: a_zeraatkar_m@yahoo.com; a_zeraatkar_m@birjand.ac.ir

Decision letter (RSOS-201076.R0)

Dear Dr Ahmad:

Title: Efficient Removal of Pharmaceuticals from Water Using Graphene Nanoplatelets as Adsorbent

Manuscript ID: RSOS-201076

The editor assigned to your manuscript has now received comments from reviewers. We would like you to revise your paper in accordance with the referee and Subject Editor suggestions which can be found below (not including confidential reports to the Editor). Please note this decision does not guarantee eventual acceptance.

Please submit your revised paper before 25-Sep-2020. Please note that the revision deadline will expire at 00.00am on this date. If we do not hear from you within this time then it will be assumed that the paper has been withdrawn. In exceptional circumstances, extensions may be possible if agreed with the Editorial Office in advance. We do not allow multiple rounds of revision so we urge you to make every effort to fully address all of the comments at this stage. If deemed necessary by the Editors, your manuscript will be sent back to one or more of the original reviewers for assessment. If the original reviewers are not available we may invite new reviewers.

Yours sincerely,
Dr Ellis Wilde
Publishing Editor, Journals

RSC Subject Editor
Comments to the Author:
(There are no comments.)

RSC Associate Editor

Comments to the Author:
(There are no comments.)

Reviewers' Comments to Author:
Reviewer: 1

Comments to the Author(s)

In this study, graphene nanoplatelets (GNPs) were used to efficiently remove antibiotics sulfamethoxazole (SMX) and analgesic acetaminophen (ACM) as pharmaceutical pollutants from water by adsorption process. Different types of GNPs; C750, C300, M15, and M5 were characterized by high-resolution transmission electron microscopy (HRTEM), Raman spectroscopy, X-Ray diffraction (XRD) and BrunauerEmmett-Teller (BET). The effects of several parameters; viz., solution pH, adsorbent amount, initial concentration and contact time were studied. The parameters were optimized by batch adsorption process and the maximum removal efficiency for both pharmaceuticals were 99%. The adsorption kinetics and isotherms were studied, and the experimental data were best analysed with pseudo-second kinetic and Langmuir isotherm model with maximum adsorption capacity (Q_m) of 210.084 mg g⁻¹ for SMX and 56.211 mg g⁻¹ for ACM. The work is interesting and well documented but needs substantial revision before the acceptance:

1. Please describe the theoretic reasons on the adsorption of SMX and ACM using graphene nanoplatelets (GNPs).
2. How about the preparation reproducibility of the graphene nanoplatelets (GNPs).
3. The characterization of any adsorbent is very important to know the surface morphology, pore size, surface area and functional groups after adsorption too. The authors must include SEM, FTIR, and BET after adsorption of SMX and ACM also and the changes after the adsorption should be explained clearly.
4. Adsorption desorption and regeneration studies must be performed for the recyclic use of the adsorbent using different eluents.
5. An English review is required in this manuscript. There are several misspelling words throughout the text. Punctuation through the text is weak, and also should be checked.
6. Why the authors focused SMX and ACM for the removal study. Did they check any adsorption efficiency towards SMX and ACM.

There are many papers published on this topic and this one follows the same structure and content as many of the previously published ones, without adding much of significance to the current knowledge. What is the novelty in this work. How the adsorption technique is better than other methods. There are some other pollutants also exist with pharmaceutical pollutants like metals and dyes which may be discussed here. The merits and demerits of adsorption technique must be added in this part. The introduction part must be improved with latest references for the superiority of adsorption for the removal of different types of pollutants. As a suggestion, following references or other similar work may be followed: Journal of molecular liquids 260, 342-350, 2018; Chemical Engineering Journal 334, 462-478, 2018; Journal of Cleaner Production 250, 119553, 2020

9. Photodegrataion is also one of the commonly use methods?? As a suggestion, the authors can follow the following work or other similar work for the discussion: Desalin Water Treat 102, 273-279, 2018; Journal of Water Process Engineering 37, 101325, 2020; Journal of hazardous materials 364, 429-440, 2019
9. Include the breakthrough studies also. Did the authors check the zeta potential also?
10. What is the practical application of the work.
11. Quality of all figures must be improved to make them clear. Improve the resolution as well as size of the figs.

Reviewer: 2

Comments to the Author(s)

Dear Authors,

The manuscript titled with "Efficient Removal of Pharmaceuticals from Water Using Graphene Nanoplatelets as Adsorbent" appears to provide a feasible adsorbent for efficient removal of pharmaceuticals from water samples. The manuscript is an interesting work and well presented, and I find this manuscript is suitable for publication in after major revision. The comments are listed below:

- 1) Please do not use the abbreviations in the abstract section.
- 2) The characterization section has detailed discussion, but FTIR analyses are suggested.
- 3) The intra-particle diffusion model study is recommended in kinetic study section.
- 4) The thermodynamic study is highly recommended.
- 5) As can be seen in many similar works which introduced new adsorbents, the desorption and reusability tests is necessary.
- 6) To investigate the applicability of the proposed new adsorbent for pollutant removal, the dye adsorption process from real samples, e.g. related industrial wastewaters, are recommended.
- 7) The discussion about the dye adsorption mechanism is not convincing. A coherent mechanism study must be done especially about the possible interactions between adsorbent and pharmaceuticals.
- 8) The authors should more explain about advantages of the proposed adsorbent and compared with the other similar adsorbent.
- 9) More explain about novelty of the work must added in the conclusion section.

Yours sincerely

Ali Zeraatkar Moghaddam,
Assistant Professor of Analytical Chemistry,
University of Birjand, Birjand, South Khorasan, Iran
E-mail address: a_zeraatkar_m@yahoo.com; a_zeraatkar_m@birjand.ac.ir

Author's Response to Decision Letter for (RSOS-201076.R0)

See Appendix A.

RSOS-201076.R1 (Revision)

Review form: Reviewer 1

Is the manuscript scientifically sound in its present form?

Yes

Are the interpretations and conclusions justified by the results?

Yes

Is the language acceptable?

Yes

Do you have any ethical concerns with this paper?

No

Have you any concerns about statistical analyses in this paper?

No

Recommendation?

Accept as is

Comments to the Author(s)

Accept.

Review form: Reviewer 2 (Ali Zeraatkar Moghaddam)

Is the manuscript scientifically sound in its present form?

Yes

Are the interpretations and conclusions justified by the results?

Yes

Is the language acceptable?

Yes

Do you have any ethical concerns with this paper?

No

Have you any concerns about statistical analyses in this paper?

No

Recommendation?

Accept with minor revision (please list in comments)

Comments to the Author(s)

Dear Authors,

The manuscript is an interesting work is suitable for publication in after minor revision.

The comments are listed below:

1- Please explain the novelty of your work in comparing with the following ref.

Doi: 10.1002/aoc.6059

2- Please added the following refs.

Doi: 10.1002/aoc.6059

Doi: 10.1016/j.ijbiomac.2019.01.086

Decision letter (RSOS-201076.R1)

Dear Dr AHMAD:

Title: Efficient Removal of Pharmaceuticals from Water Using Graphene Nanoplatelets as Adsorbent

Manuscript ID: RSOS-201076.R1

Thank you for submitting the above manuscript to Royal Society Open Science. On behalf of the Editors and the Royal Society of Chemistry, I am pleased to inform you that your manuscript will be accepted for publication in Royal Society Open Science subject to minor revision in accordance with the referee suggestions. Please find the reviewers' comments at the end of this email.

The reviewers and handling editors have recommended publication, but also suggest some minor revisions to your manuscript. Therefore, I invite you to respond to the comments and revise your manuscript.

Because the schedule for publication is very tight, it is a condition of publication that you submit the revised version of your manuscript before 03-Dec-2020. Please note that the revision deadline will expire at 00.00am on this date. If you do not think you will be able to meet this date please let me know immediately.

Kind regards,
Dr Laura Smith
Publishing Editor, Journals

RSC Associate Editor:
Comments to the Author:
(There are no comments.)

RSC Subject Editor:
Comments to the Author:
(There are no comments.)

Reviewer comments to Author:
Reviewer: 1

Comments to the Author(s)
Accept.

Reviewer: 2

Comments to the Author(s)
Dear Authors,
The manuscript is an interesting work is suitable for publication in after minor revision.
The comments are listed below:
1- Please explain the novelty of your work in comparing with the following ref.
Doi: 10.1002/aoc.6059
2- Please added the following refs.
Doi: 10.1002/aoc.6059
Doi: 10.1016/j.ijbiomac.2019.01.086

Author's Response to Decision Letter for (RSOS-201076.R1)

See Appendix B.

Decision letter (RSOS-201076.R2)

Dear Dr AHMAD:

Title: Efficient Removal of Pharmaceuticals from Water Using Graphene Nanoplatelets as Adsorbent
Manuscript ID: RSOS-201076.R2

It is a pleasure to accept your manuscript in its current form for publication in Royal Society Open Science. The chemistry content of Royal Society Open Science is published in collaboration with the Royal Society of Chemistry.

RSC Associate Editor
Comments to the Author:
(There are no comments.)

Reviewer(s)' Comments to Author:

Appendix A

RESPONSE TO REVIEWERS

Title: Efficient Removal of Pharmaceuticals from Water Using Graphene Nanoplatelets as Adsorbent

Manuscript ID: RSOS-201076

Dear Professor Anthony Stace and Dr Ya-Wen Wang,

Thank you for giving us the opportunity to submit a revised draft of the manuscript "Efficient Removal of Pharmaceuticals from Water Using Graphene Nanoplatelets as Adsorbent" for publication in the Royal Society of Open Science. We appreciate the time and effort that you and the reviewers dedicated to providing feedback on our manuscript and are grateful for the insightful comments on and valuable improvements to our paper. We have incorporated most of the suggestions made by the reviewers. Please see below, in green, for a point-by-point response to the reviewers' comments and concerns. Some of the suggestions we cannot afford to fulfil due to Movement Control Order in our country, Malaysia. We have tried as best as we could to fulfil the reviewer's comments and hopefully it meets your expectation. We hope the revised manuscript will better suit the Royal Society of Open Science and we thank you for your continued interest in our research.

Sincerely,

Haslina Ahmad

Reviewer 1

Comments to the Author(s):

In this study, graphene nanoplatelets (GNPs) were used to efficiently remove antibiotics sulfamethoxazole (SMX) and analgesic acetaminophen (ACM) as pharmaceutical pollutants from water by adsorption process. Different types of GNPs; C750, C300, M15, and M5 were characterized by high-resolution transmission electron microscopy (HRTEM), Raman spectroscopy, X-Ray diffraction (XRD) and BrunauerEmmett-Teller (BET). The effects of several parameters; viz., solution pH, adsorbent amount, initial concentration and contact time were studied. The parameters were optimized by batch adsorption process and the maximum removal efficiency for both pharmaceuticals were 99%. The adsorption kinetics and isotherms were studied, and the experimental data were best analysed with pseudo-second kinetic and Langmuir isotherm model with maximum adsorption capacity (Q_m) of 210.084 mg g⁻¹ for SMX and 56.211 mg g⁻¹ for ACM. The work is interesting and well documented but needs substantial revision before the acceptance:

1. Please describe the theoretic reasons on the adsorption of SMX and ACM using graphene nanoplatelets (GNPs).

Author response: Thank you for this excellent suggestion. We made some changes to emphasize this in the manuscript. "...due to its enormous, delocalized π - π electron system thus can form strong bonding with various pollutants.."(page 2) "Both SMX and ACM molecules contain aromatic group or benzene ring.." "The adsorption mechanism mainly due to interaction of the electron rich GNP surface and the protonated aniline ring of pharmaceutical molecules.." (page 12)

2. How about the preparation reproducibility of the graphene nanoplatelets (GNPs).

Author response: Thank you for pointing this out. As mentioned in page 2, the graphene nanoplatelets were purchased directly from XG Sciences, USA which produces high quality GNPs.

3. The characterization of any adsorbent is very important to know the surface morphology, pore size, surface area and functional groups after adsorption too. The authors must include SEM, FTIR, and BET after adsorption of SMX and ACM also and the changes after the adsorption should be explained clearly.

Author response: As suggested by the reviewer, we have added FESEM/EDX, BET for the changes after adsorption in page 11-12. Unfortunately, we could not get the FTIR results since we cannot resume our laboratory work due to restriction of movement order from our government.

4. Adsorption desorption and regeneration studies must be performed for the recyclic use of the adsorbent using different eluents.

Author response: Thank you for pointing this out. We have added the suggested content to the manuscript in page 12. "As depicted in Figure 11, the removal efficiency for both SMX and ACM using GNP C300 through four regeneration cycles by different eluents; 0.005 M NaOH, 0.005 M HCl and 5% ethanol-deionized water...."

5. An English review is required in this manuscript. There are several misspelling words throughout the text. Punctuation through the text is weak, and also should be checked.

Author response: Thank you so much for catching these errors, which we have now corrected.

6. Why the authors focused SMX and ACM for the removal study. Did they check any adsorption efficiency towards SMX and ACM. There are many papers published on this topic and this one follows the same structure and content as many of the previously published ones, without adding much of significance to the current knowledge. What is the novelty in this work. How the adsorption technique is better than other methods. There are some other pollutants also exist with pharmaceutical pollutants like metals and dyes which may be discussed here. The merits and demerits of adsorption technique must be added in this part. The introduction part must be improved with latest references for the superiority of adsorption for the removal of different types of pollutants. As a suggestion, following references or other similar work may be followed: Journal of molecular liquids 260, 342-350, 2018; Chemical Engineering Journal 334, 462-478, 2018; Journal of Cleaner Production 250, 119553, 2020

Author response: Thank you for the review. The removal percentage/efficiency is the same as adsorption efficiency. The amount of adsorbed pollutants reflected the percentage of pollutants have been removed. The advantages for adsorption technique have been emphasized in page 2. Previous research only focuses on C750 since it has the highest surface area. In this paper we compared four different types of GNP and C300 turns out to work better than C750. (page 2 & 13) "The unexplored potential of GNP C300 and the rest has revealed the other types GNP that works better compared to C750 which is a bit costly and facing agglomeration problem." Thank you for the references.

7. Photodegradation is also one of the commonly use methods?? As a suggestion, the authors can follow the following work or other similar work for the discussion: Desalin Water Treat

102, 273-279, 2018; Journal of Water Process Engineering 37, 101325, 2020; Journal of hazardous materials 364, 429-440, 2019

Author response: We appreciate the suggestion and the photodegradation method also included together as well with degradation method (page 2). Thank you for the references.

8. Include the breakthrough studies also. Did the authors check the zeta potential also?

Author response: Thank you for the suggestion. We have added this in page 7.

9. What is the practical application of the work.

Author response: Thank you for pointing this out. We have added the application towards environmental water samples (page 13).

10. Quality of all figures must be improved to make them clear. Improve the resolution as well as size of the figs.

Author response: Thank you. We already change the resolution of the figures for clearer view.

Reviewer: 2

Comments to the Author(s)

Dear Authors,

The manuscript titled with "Efficient Removal of Pharmaceuticals from Water Using Graphene Nanoplatelets as Adsorbent" appears to provide a feasible adsorbent for efficient removal of pharmaceuticals from water samples. The manuscript is an interesting work and well presented, and I find this manuscript is suitable for publication in after major revision. The comments are listed below:

1) Please do not use the abbreviations in the abstract section.

Author response: Thank you for pointing this out. The reviewer is correct, and we have removed the abbreviations and replaced them.

2) The characterization section has detailed discussion, but FTIR analyses are suggested.

Author response: Thank you for this suggestion. It would have been interesting to explore this aspect. However, we did not have enough time to complete FTIR study as the government has ruled out a local lock down recently.

3) The intra-particle diffusion model study is recommended in kinetic study section.

Author response: Thank you for this suggestion. We have added this part in Page 10.

4) The thermodynamic study is highly recommended.

Author response: Thank you for pointing this out. Although we agree that this is an important consideration, this study cannot resume due to local lock down.

5) As can be seen in many similar works which introduced new adsorbents, the desorption and reusability tests is necessary.

Author response: Thank you for pointing this out. We have added the suggested content to the manuscript in page 12-13.

6) To investigate the applicability of the proposed new adsorbent for pollutant removal, the dye adsorption process from real samples, e.g. related industrial wastewaters, are recommended.

Author response: Thank you for this suggestion. We have added the application towards environmental water samples; Kuyoh river water, Langat river water and tap water (page 13).

7) The discussion about the dye adsorption mechanism is not convincing. A coherent mechanism study must be done especially about the possible interactions between adsorbent and pharmaceuticals.

Author response: As suggested by the reviewer, we have added FESEM/EDX, BET for the changes after adsorption in page 11-12. On top of that, molecular docking also included as well. We also add some discussion for the possible interactions for the adsorption mechanism.

8) The authors should more explain about advantages of the proposed adsorbent and compared with the other similar adsorbent.

Author response: We think this is an excellent suggestion. As suggested, we have added comparison with similar graphitic and carbonaceous adsorbents in the manuscript (page 11).

9) More explain about novelty of the work must added in the conclusion section.

Author response: We agree with the reviewer's assessment. Accordingly, throughout the manuscript, we have revised the conclusion section (page 13-14).

Appendix B

RESPONSE TO REVIEWERS

Title: Efficient Removal of Pharmaceuticals from Water Using Graphene Nanoplatelets as Adsorbent

Manuscript ID: RSOS-201076.R1

Dear Professor Anthony Stace, Dr Ya-Wen Wang and Dr Laura Smith,

Thank you for giving us the opportunity to submit a revised draft of the manuscript "Efficient Removal of Pharmaceuticals from Water Using Graphene Nanoplatelets as Adsorbent" for publication in the Royal Society of Open Science. We appreciate the time and effort that you and the reviewers dedicated to providing feedback on our manuscript and are grateful for the insightful comments on and valuable improvements to our paper. We thank the reviewer for the kind comment and useful suggestions for which we have used to improve our manuscript. We have incorporated most of the suggestions made by the reviewers. Please see below, in green, for a point-by-point response to the reviewers' comments and concerns. We hope the revised manuscript will better suit the Royal Society of Open Science and we thank you for your continued interest in our research.

Sincerely,

Haslina Ahmad

RSC Associate Editor:

Comments to the Author:

(There are no comments.)

RSC Subject Editor:

Comments to the Author:

(There are no comments.)

Reviewer comments to Author:

Reviewer: 1

Comments to the Author(s)

Accept.

Reviewer: 2

Comments to the Author(s)

Dear Authors,

The manuscript is an interesting work is suitable for publication in after minor revision.

The comments are listed below:

1- Please explain the novelty of your work in comparing with the following ref.

Doi: 10.1002/aoc.6059

Author response: In the following reference, iron modified graphene oxide was used for the degradation of tetracycline antibiotic. It was reported that the maximum removal efficiency is between 73.9 - 88.8%. Meanwhile, in our research, graphene nanoplatelet (GNP) C300 was used as the adsorbent and we reported the removal of antibiotics sulfamethoxazole (SMX) and analgesic acetaminophen (ACM) as pharmaceutical pollutants from water by adsorption process. We observed the maximum removal efficiency for both pharmaceuticals (99%). We also applied the adsorbent to remove the pharmaceutical pollutants from three water samples (tap water, Kuyoh river water and Langat river water). The adsorbent used in this research also able to achieve high removal efficiencies even after the fourth cycle of regeneration process for both pharmaceutical pollutants. In addition, the Brunauer-Emmett-Teller (BET) surface area of GNP C300 ($390.3 \text{ m}^2 \text{ g}^{-1}$) used in this study is much higher compared to GO@Fe/Cu/Ag ($77 \text{ m}^2 \text{ g}^{-1}$).

2- Please added the following refs.

Doi: 10.1002/aoc.6059

Doi: 10.1016/j.ijbiomac.2019.01.086

Author response: Thank you for the suggestion, the corresponding references have been added in the manuscript.

“There are various removal methods that are used to remove pollutants from water such as bioreactors [14], degradation and photodegradation [15–18] ...” (please refer page 2)

and

“This technique is superior considered as powerful alternatives to conventional methods [26] because of its low cost, ease of operation...” (please refer page 2)